# Information Entropy-Based Intention Prediction of Aerial Targets under Uncertain and Incomplete Information

**DOI:** 10.3390/e22030279

**Published:** 2020-02-28

**Authors:** Tongle Zhou, Mou Chen, Yuhui Wang, Jianliang He, Chenguang Yang

**Affiliations:** 1College of Automation Engineering, Nanjing University of Aeronautics and Astronautics, Nanjing 211106, China; zhoutongle@nuaa.edu.cn (T.Z.); wangyh@nuaa.edu.cn (Y.W.); 2Science and Technology on Electro-Optic Control Laboratory, Luoyang 471000, China; 3Bristol Robotics Laboratory, University of the West of England, Bristol BS16 1QY, UK; cyang@ieee.org

**Keywords:** state prediction, LSTM networks, intention recognition, decision tree, data missing, interval-valued

## Abstract

To improve the effectiveness of air combat decision-making systems, target intention has been extensively studied. In general, aerial target intention is composed of attack, surveillance, penetration, feint, defense, reconnaissance, cover and electronic interference and it is related to the state of a target in air combat. Predicting the target intention is helpful to know the target actions in advance. Thus, intention prediction has contributed to lay a solid foundation for air combat decision-making. In this work, an intention prediction method is developed, which combines the advantages of the long short-term memory (LSTM) networks and decision tree. The future state information of a target is predicted based on LSTM networks from real-time series data, and the decision tree technology is utilized to extract rules from uncertain and incomplete priori knowledge. Then, the target intention is obtained from the predicted data by applying the built decision tree. With a simulation example, the results show that the proposed method is effective and feasible for state prediction and intention recognition of aerial targets under uncertain and incomplete information. Furthermore, the proposed method can make contributions in providing direction and aids for subsequent attack decision-making.

## 1. Introduction

In modern air combat, the vigorous development of aviation science and military technology leads to more and more severe threats of aerial targets. Meanwhile, due to the application of high-tech technology of Unmanned Combat Air Vehicle (UCAV), such as space early warning systems, radar stealthy composites, artificial intelligence technology, etc., the complexity of a battlefield environment including uncertainty and incompleteness is increasing [1]. Therefore, predicting target state and recognizing intention previously can favor making adequate preparations for air combat autonomous attack and defense decision-making systems. Furthermore, state prediction and intention recognition of aerial targets also have great contributions on increasing the operational efficiency of weapon systems and saving the resources of air combat. Hence, state prediction and intention recognition of aerial targets are important sections of air combat decision support systems, and they have been playing vital roles in future command and control systems [2].

In recent years, many decision support research results have been carried out on military fields to satisfy the requirement of combat decision-making systems. In [3], a rough set theory-based multi-criteria decision-making (MCDM) model was proposed, in which the exceptional importance of the software application to decision-making in the security forces operations was demonstrated. In [4], a hybrid MCDM model in the determination and evaluation of the criteria for selecting an aircraft was presented for the protection of air traffic. A decision support system is an enabling technology leading to numerous disruptive changes in the military field. Using decision support systems reasonably and effectively can greatly enhance the competitiveness in combat decision-making systems. As important parts of decision support systems, state prediction and intention recognition technology can make use of detected information to reflect the actual situation and lay a foundation for decision-making in air combat. Some works can be found in related literature. In [5], a novel method based on support vector machine and Bayesian filtering was studied for online lane change intention prediction in road vehicle driving. To predict the air combat data effectively and accurately, a target state prediction method was introduced in [6] for an aerial target based on the autoregressive integrated moving average (ARIMA) model. An intention prediction method was studied for the aerial target based on an improved grey incidence analysis method in [7]. An algorithm for assessment of the target maneuvering intention in beyond-visual-range air-combat was proposed in [8], in which the target maneuvering intention was divided into nine categories and the characteristic parameters of the target were extracted according to target real-time data being measured and predicted; then the threat level and maneuvering intention could be estimated. Aiming at the difficulty to quantify the mapping relationship between attribute features and combat intentions under the condition of insufficient knowledge of domain experts, a method of combat intention recognition based on deep neural networks was proposed in [9]. In [10], a self-learning method based on decision tree was studied to solve naval vessel intention recognition problem. Although state prediction and intention recognition problems of aerial targets have been studied in recent years, the uncertainty or incompleteness, especially the condition of existing simultaneously in air combat environments is seldom discussed in existing research results.

Long short-memory term (LSTM) network is an improvement over the general recurrent neural network (RNN) [11]. Unlike traditional RNNs, LSTM networks are suitable for learning from experience to classify, process and predict time series when there are very long time lags of unknown size between important events [12,13,14,15,16]. Based on the recent success of LSTM networks for time series domains, a convolutional and LSTM recurrent units-based deep framework was proposed in [17] for activity recognition. To predict the time series of traffic and user mobility in telecommunication networks, a random connectivity LSTM model was put forward in [18]. In air combat, the state data of a target is also a kind of time series. Thus, to solve the aerial target state prediction problem, LSTM networks are employed in this paper. After state prediction, a decision tree is used to extract the rules from the uncertain and incomplete historical data, then the intention of an ariel target is calculated based on the predicted state data and decision tree classification rules. As a decision support tool, a decision tree employs a tree-like graph or model of the decision and its possible consequences, containing chance event outcomes, resource costs and utility [19]. In addition, a decision tree is widely applied in operations research, particularly in decision-making analysis, to help determine a strategy most probable to achieve a goal, but is also a useful tool in machine learning [20,21,22].

In this paper, to solve the difficulties of aerial target intention prediction with uncertainty and incompleteness, the state prediction approach and intention recognition methods are designed synthetically. Firstly, the target state data is predicted based on the LSTM networks from the real-time series data collected through multi-sensors. Then, the improved decision tree theory is used to extract the rules from historical data and is applied to handle the information system with uncertainty and incompleteness. Finally, the intention is recognized by inputting the predicted state data to the built decision tree.

In the rest of the paper, the air combat situation is illustrated, then the definition of air combat uncertainty and incompleteness are elaborated in Section 2. In Section 3, the state prediction method based on LSTM networks is presented. Then, the intention recognition decision tree is generated with the uncertain and incomplete historical data based on information entropy, which is displayed in Section 4. The simulation results are shown in Section 5. The conclusion is summarized in Section 6.

## 2. Problem Statement

In the modern air combat environment with uncertainty and incompleteness, the air combat data consists of real-time data and priori knowledge [23]. The real-time data is obtained through various kinds of sensors in the air combat process. As to the priori knowledge, it is the historical information and rules in the past air combat. The aerial target state information can be predicted according to the air combat situation and the motion properties of the target. The future state data can be obtained through the analysis of the real-time data in air combat. Furthermore, the intention recognition rules are acquired by the means of priori knowledge and the intention can be recognized.

In this paper, the intention of a target is divided into attack, surveillance, penetration, feint, defense, reconnaissance, cover and electronic interference. Because of the regularity of aerial target intention, different target states can reflect the various results. For example, the high speed targets are more aggressive with a larger possibility of attack intention when the target sight direction towards the UCAV.

The air combat situation diagram between the target and UCAV is shown in Figure 1 [24].

In Figure 1, the line between the target and UCAV is the target line of sight. *A* is the angle between target line and due north, which is called the azimuth of a target. *D* is the distance between target and UCAV. *V* is the velocity of the target and Ha is the heading angle of target, the angle between target velocity and target line of sight. For convenience, the *H* means the height difference in this paper. Obviously, the air combat situation factors are numerical data.

Apart from the above state factors of a target, the intention of a target is also related to the operational task supplementary of the target, such as air-to-air radar status, marine radar status, disturbing state and disturbed state. These factors of operational task are nonnumerical data lies on a nominal scale, which can be expressed as 0 and 1. For instance, the range of air-to-air radar status and marine radar status are {0,1}, where 0 represents that the radar is on the off state and 1 represents that the radar is on the open state.

Thus, all major factors of air combat situations and operational task supplementary shall be taken into consideration, from which other factors can be derived. The advantage is that the state of UCAV is not indispensable. The tree chart of target intention characteristic description in air combat is shown in Figure 2.

The objective of this paper consists of two parts. The first part is to design a target state prediction algorithm based on real-time data. The second part is to extract rules from the priori knowledge, and identify the intention in accordance with the predicted state data.

The state prediction and intention recognition system diagram is shown in Figure 3.

It should be noted that the priori knowledge maybe uncertain and incomplete due to the complexity of air combat and the confusability of the target. The uncertainty and incompleteness are defined as follows:**Uncertainty**: The specified value of aerial target state is hard to obtain accurately because of the limitation of sensors and rapidity of air combat. In such cases, only a specific range can be detected by multi-sensors. Hence, some state information is expressed as interval-valued number in this paper to describe the uncertainty of air combat.**Incompleteness**: In the air combat process, some information of a target may can not be detected due to the application of innovative military technology. In addition, missing data may happen to historical data. Therefore, the priori knowledge is incomplete.

## 3. State Prediction based on LSTM Networks

LSTM network is a recurrent neural network (RNN) architecture introduced by Hochreiter and Schmidhuber in 1997. Compared with traditional RNNs, LSTM networks contain four interacting layers, which are called cell state, forget gate, input gate and output gate [24]. According to the knowledge of [25], the structure of LSTM networks is shown as follow:

In Figure 4, $C_{t}$ and $C_{t-1}$ are the cell states, $h_{t}$ and $h_{t-1}$ are the hidden layer states and $x_{t}$ is the input.

Obviously, the key of LSTM networks is the self-connected memory cell state, the horizontal line running through the top of the LSTM networks structure. The LSTM networks can add or delete information to the cell state based on the structures called gates. Under the control of gates, the information is passed optionally in the cell state. In LSTM networks, the gates consist of a sigmoid neural net layer $\sigma_{g}\left(x\right)=\frac{1}{1+e^{-x}}$ and a pointwise multiplication operation [25].

Firstly, the forget gate layer decides the deleted information from the cell state. From Figure 4, the output of a forget gate can be expressed as [25]
(1)$$\begin{array}{c} f_{t}=\sigma_{g}\left(W_{f}\cdot \left[h_{t-1}\;x_{t}\right]+b_{f}\right) \end{array}$$
where $W_{f}$ is the input weight matrix and $b_{f}$ is the bias weight matrix of forget gate layer. $\left[h_{t-1}\;x_{t}\right]$ is a vector.

If $f_{t}=1$, the information of $h_{t}$ and $x_{t}$ is retained completely, while $f_{t}=0$ means completely losing the information. Namely, the greater $f_{t}$ means the more information is retained.

The next step of LSTM networks is to determine what new information will be stored in the cell state. This step has two parts. Firstly, which values will be updated is decided by the input layer. Then, the output of input gate $i_{t}$ can be expressed as [25]
(2)$$\begin{array}{c} i_{t}=\sigma_{g}\left(W_{i}\cdot \left[h_{t-1}\;x_{t}\right]+b_{i}\right) \end{array}$$
where $W_{i}$ is the input weight matrix and $b_{i}$ is the bias weight matrix of input gate layer.

Next, a tanh layer generates a vector of new candidate values $\tilde{C_{t}}$, we have [25]
(3)$$\begin{array}{c} \tilde{C_{t}}=\tan h\left(W_{C}\cdot \left[h_{t-1}\;x_{t}\right]+b_{C}\right) \end{array}$$
where $W_{C}$ is the input weight matrix and $b_{C}$ is the bias weight matrix of cell state.

Then, these two parts will be combined to create an update to the cell state and obtain $C_{t}$. We have [26]
(4)$$\begin{array}{c} C_{t}=f_{t}\cdot C_{t-1}+i_{t}\cdot \tilde{C_{t}} \end{array}$$

Finally, what LSTM networks is going to output $o_{t}$ should be confirmed. The output gate layer decides the output parts of the cell state. It can be expressed as [25]
(5)$$\begin{array}{c} o_{t}=\sigma_{g}\left(W_{o}\cdot \left[h_{t-1}\;x_{t}\right]+b_{o}\right) \end{array}$$
where $W_{o}$ is the input weight matrix and $b_{o}$ is the bias weight matrix of output gate layer.

Then, LSTM networks put the cell state through tanh function and multiply it by the output of the sigmoid layer. Thus, LSTM networks only output the selected parts [26].
(6)$$\begin{array}{c} h_{t}=o_{t}\cdot \tan h\left(C_{t}\right) \end{array}$$

In general, the real-time numeric data of a target is essentially a time series which can be expressed as f(1), f(2), …, f(t). The function of LSTM networks is predicting f(t+1) based on f(1), f(2), …, f(t).

To solve the lack of the training data problem in real-time air combat, suppose that there are *N* time-lagged observations f(1), f(2), …, f(N) in the training set and we need the one-step-ahead prediction. In order to avoid the problem of limited training sample and improve the predict accuracy, a network with *p* input nodes and one output node is used in this section. Hence, we have N−p training patterns.

Assume the size of the time window is *p*, the first training pattern is composed of f(1), f(2), …, f(p) as the inputs and f(p+1) as the target output. The second training pattern is composed of f(2), f(3), …, f(p+1) as the inputs and f(p+2) as the target output. By that analogy, the last training pattern is f(N−p−1), f(N−p), …, f(N−1) for the inputs and f(N) for the target output. The additional benefit is that we can make full use of the real-time data to train the LSTM networks. So far, the aerial target state prediction training model is established. Finally, f(N−p), f(N−p+1), …, f(N) are chosen as the input pattern, and the output f(N+1) is the predicted state data.

For nonnumeric data of factors, the observations is expressed as $f_{n}\left(1\right)$, $f_{n}\left(2\right)$, …, $f_{n}\left(t\right)$. Due to the high-performance early warning radar, we assume that the operational task supplementary information of target is same as last time. Namely, we have
(7)$$\begin{array}{c} f_{n}\left(N+1\right)=f_{n}\left(N\right) \end{array}$$
where $f_{n}\left(t\right)\left(t=1,2,\ldots ,N\right)$ are the nonnumeric observations of target operational task supplementary.

The structure of aerial target state prediction is shown as Figure 5:

As mentioned above, the state factors and operational task supplementary are chosen as the intention features in this paper. Hence, for each feature, the predicted value can be obtained by the prediction model, and the next step is intention recognition.

## 4. Target Intention Recognition Based on Decision Tree and Information Entropy

After the state has been predicted, intention recognition is considered according to the predicted state information. Hence, intention recognition is also an important part of an air combat decision-making system of UCAV.

Because of the incompleteness and the uncertainty of air combat, it is hard to extract the rules from historical data. In this paper, the incompleteness is denoted as the missing data (null value) that may exist in historical data, and the uncertainty can be expressed as interval number. The purpose is to build a decision tree based on an incomplete and interval-valued historical information decision table. Moreover, the input of this part is the predicted value of a target state prediction model, and the output is the target intention.

A decision tree is a flowchart-like structure, in which each internal node represents a test of an attribute, each branch represents the output of the test, and each leaf node represents a class label [27]. The paths from root to leaf represent classification rules. A decision tree has been widely used in classification, information retrieval and dimensionality reduction, and there are broad prospects for development. It can be trained in either supervised or unsupervised ways, depending on the task.

The two major problems in building a decision tree are node splitting order selection and how to choose the best split criterion of nodes. In this paper, a decision support degree is applied for node splitting order selection and split criterion is determined by the information entropy of partitioning.

The structure of decision tree generation for aerial target intention recognition is shown as Figure 6:

As is well known, the priori knowledge of air combat S=(U,A∪D) is a kind of incomplete information system, where *U* is a finite nonempty set of statistics objects of historical data. *A* is a finite nonempty set of condition attributes namely, the threat factors, and *D* is a finite nonempty set of decision attributes, namely, the intention of aerial target in historical data. $\exists a_{i}\in A,a_{i}=*$, where the special symbol “*” denotes that the value of an attribute is unknown. On the other hand, $\forall d_{i}\in D,d_{i}\neq *$.

To handle the incomplete information, the following existing definitions are needed.

**Definition** **1.**
*Let*
S=(U,A∪D)
*be an interval-valued attributes based incomplete system, where A is the set of condition attributes, D is the set of decision attributes and*
∗∉D
*. A similarity relation*
SIM(R)(R⊆A)
*on U is defined as follows [28]*
(8)$$\begin{array}{c} SIM(R)=\{(u,v)\in U\times U|\forall a\in A,f(u,a)=f(v,a)\;or\;f(u,a)=*\;or\;f(v,a)=*\} \end{array}$$
*where*
f:U→A
*is the mapping from U to A.*


According to the definition of SIM(R), if (u,v)∈U×U are in SIM(R), they are perceived as similar. Namely, they may have the same properties with respect to *R* in reality.

In the similarity relation SIM(R), ∀u∈U, Let $S_{p}\left(U\right)=\left\{\forall v\in U|\left(u,v\right)\in SIM\left(R\right)\right\}$, where $S_{p}\left(U\right)$ is called the consistent block of *U*. In other words, $S_{p}\left(U\right)$ is the maximizing set of indistinguishable object [24].

Actually, the process of intention recognition is extracting rules from historical knowledge and identifying the intention based on the predicted value. On the basis of this, building an incomplete decision tree is a classification issue. In the literature, the guessing technologies are often used in the building of an incomplete system decision tree [29]. In this paper, a condition attribute decision support degree with respect to the decision attribute is defined as follow:

**Definition** **2.**
*Let*
S=(U,A∪D)
*be an interval-valued attributes based incomplete system, where A is the set of condition attributes, D is the set of decision attributes and*
∗∉D
*.*
R⊆A
*,*
$U/R=R_{1},R_{2},\ldots ,R_{m}$
*,*
$U/D=D_{1},D_{2},\ldots D_{n}$
*, let*
$\left|U/R\right|=\sum_{i=1}^{m}\left|R_{i}\right|$
*, the decision support degree*
DSD(R,D)
*of condition attribute R to decision attribute D is defined as [30]*
(9)$$\begin{array}{c} DSD\left(R,D\right)=1-\frac{\sum_{i=1}^{m}\sum_{j=1}^{n}\left|R_{i}\cap D_{j}|\times |R_{i}-D_{j}\right|}{|U/R|\times (|U|-1)-\sum_{l=1}^{n}\left(|D_{l}|\times (|D_{l}|-1)\right)} \end{array}$$


Decision support degree indicates the support level of condition attribute *R* to partition U/R. The large value of decision support degree lies in the better effect of classification based on condition attribute *R*. Thus, the attribute with greater decision support degree should be split preferentially.

For air combat incomplete interval-valued information system, the set of condition attributes is
(10)$$\begin{array}{c} A_{ac}=\left\{A,D,V,Ha,H,Ars,Mrs,Ds,Dds\right\} \end{array}$$
where *A*, *D*, *V*, Ha, *H*, Ars, Mrs, Ds and Dds are the azimuth, distance, velocity, heading angle, height, air-to-air radar status, marine radar status, disturbing state and disturbed state of target. In $A_{ac}$, $\forall a_{i}\in A_{ac}$, $a_{i}=*$ or $a_{i}=\left[a_{i}^{L},a_{i}^{U}\right]$, $a_{i}^{L},a_{i}^{U}\in R$ and $a_{i}^{L}\leq a_{i}^{U}$, $a_{i}^{L},a_{i}^{U}$ are the endpoints of interval number $a_{i}$.

If the set of decision attributes is the intention set, we have
(11)$$\begin{array}{c} D_{ac}=\left\{A,S,P,F,D,R,C,E\right\} \end{array}$$
where *A*, *S*, *P*, *F*, *D*, *R*, *C* and *E* express attack, surveillance, penetration, feint, defense, reconnaissance, cover and electronic interference.

For numeric data in $A_{ac}$, we define
(12)$$\begin{array}{c} \begin{array}{c} V_{A}=\left\{East,South,West,North\right\}\\ V_{D}=\left\{Short,Medium,Long\right\}\\ V_{V}=\left\{Slow,Medium,Fast\right\}\\ V_{Ha}=\left\{Small,Medium,Large\right\}\\ V_{H}=\left\{Low,Medium,High\right\} \end{array} \end{array}$$
where $V_{A}$, $V_{D}$, $V_{V}$, $V_{Ha}$ and $V_{H}$ are the ranges of azimuth, distance, velocity, heading angle and height.

For nonnumeric data in $A_{ac}$, we define
(13)$$\begin{array}{c} \begin{array}{c} V_{Ars}=\left\{0,1\right\}\\ V_{Mrs}=\left\{0,1\right\}\\ V_{Ds}=\left\{0,1\right\}\\ V_{Dds}=\left\{0,1\right\} \end{array} \end{array}$$

In (13), $V_{Ars}$ and $V_{Mrs}$ are the ranges of air-to-air radar status and marine radar status, 0 represents that the radar is on the off state and 1 represents that the radar is on the open state. $V_{Ds}$ is the range of disturbing state, 0 means that the target is not jamming the UCAV and 1 means that the target is on the jamming state. $V_{Dds}$ is the ranges of disturbed state, 0 means that the target is not being jammed by UCAV and 1 means that the target is being jammed by UCAV.

Therefore, the first step is to determine the condition attribute class of interval number $a_{i}=\left[a_{i}^{L},a_{i}^{U}\right]\in A_{ac}$. The fuzzy inference is used to solve this problem. As an online decision support tool, fuzzy inference theory has contributed to achieve classification tasks, process simulation and diagnosis and process control [31].

Take the $a_{i}^{*}$ as the representative point of the interval $\left[a_{i}^{L},a_{i}^{U}\right]$, which is given by
(14)$$\begin{array}{c} a_{i}^{*}=\frac{\int_{a_{i}^{L}}^{a_{i}^{U}}y\mu_{i}\left(y\right)dy}{\int_{a_{i}^{L}}^{a_{i}^{U}}\mu_{i}\left(y\right)dy} \end{array}$$
where $\mu_{i}\left(y\right)$ is the membership function of attribute *i*.

Then, the condition attribute class of $a_{i}$ is obtained by the membership function and $a_{i}^{*}$, the membership functions of azimuth, distance, velocity, heading angle and height are designed as Figure 7, Figure 8, Figure 9, Figure 10 and Figure 11:

In this way, the air combat incomplete interval-valued information system is converted to a traditional information system and the node splitting order can be selected by the decision support degree.

The next step is to determine the best split criterion (selecting the cutpoint) of nodes, and this part is required to consider the air combat incomplete interval-valued information system renewedly.

**Definition** **3.**
*[30]: **Cutpoint**, for an interval-valued condition attribute*
$a=[a^{L},a^{U}]$
*(finite interval), cutpoint is the threshold*
$C(a^{L}\leq C\leq a^{U})$
*which splits the interval-value condition attribute a into two branches (*
$a_{1}=\left[a^{L},C\right]$
*and*
$a_{2}=\left[C,a^{U}\right]$
*).*


It is supposed that condition attribute *B* of an air combat state is chosen to split. The nonempty elements is $b_{i}=\left[b_{i}^{L},b_{i}^{U}\right]\in A_{ac}\left(i=1,2,\ldots ,M\right)$, where *M* is the number of the attribute sample. Then the sequence of 2M points can be obtained by sorting the endpoints of $b_{i}$ in ascending order. Delete the repetitive endpoints and the midpoints of each two neighbor points of the sequence are defined as the alternative cutpoints. The objective is to select the optimal cutpoint from the alternative cutpoints and it is determined by the information entropy of partitioning.

Assume that the decision attribute of selected condition attribute *B* is $D_{B}=\left\{D_{1},D_{2},\ldots ,D_{k}\right\}$, where *k* is the classes number of decision attribute. The information entropy of selected condition attribute *B* is defined as [32]
(15)$$\begin{array}{c} I\left(B\right)=-\sum_{j=1}^{k}\frac{\left|D_{j}\right|}{\left|D_{B}\right|}\log\frac{\left|D_{j}\right|}{\left|D_{B}\right|} \end{array}$$
where |·| expresses the number of elements in a set.

As the selected condition attribute *B* can be divided into two subset $B_{1}$ and $B_{2}$ by an alternative cutpoint *C*, where $C>B_{1}$ and $C\leq B_{2}$. The information entropy of partitioning IEP(B,C) is defined as [33]
(16)$$\begin{array}{c} IEP\left(B,C\right)=\frac{\left|B_{1}\right|}{\left|B\right|}\cdot I\left(B_{1}\right)+\frac{\left|B_{2}\right|}{\left|B\right|}\cdot I\left(B_{2}\right) \end{array}$$

If one of the alternative cutpoint $C^{*}$ makes the $IEP(B,C^{*})$ minimums among all alternative cutpoints of selected condition attribute *B*, the alternative cutpoint $C^{*}$ is marked as optimal cutpoint.

Finally, the decision tree is generated by the decision support degree-based node splitting order and split according to the optimal cutpoint of each condition attribute until all the non-empty elements of node are belong to the same class.

The whole decision tree generation algorithm is summarized in Algorithm 1.

At this point, the rules extraction is accomplished from the uncertain and incomplete priori knowledge, and the future aerial target intention can be recognized on the basis of predicted state data from LSTM networks and the generated decision tree.
**Algorithm 1** Decision tree generation algorithms in air combat incomplete interval-valued information system.**Input:** The air combat incomplete interval-valued information decision table of priori knowledge.**Output:** A decision tree.1:Determine the condition attribute classes of all interval number based on fuzzy inference;2:Generate a fuzzy incomplete decision table;3:**for** each episode **do**4: Calculate the decision support degree according (9) of each condition attribute;5: Choose the attribute with maximum decision support degree as the split node;6: Count the alternative cutpoints of selected condition attribute;7: Calculate the information entropy of partitioning of each alternative cutpoint based on (15)8:and (16);9: Choose the alternative cutpoint with minimum information entropy of partitioning as the10:optimal cutpoint to split the condition attribute;11: Delete the split condition attribute from incomplete interval-valued information decision table12:and fuzzy incomplete decision table.13:**until** All the condition attributes are split and all the non-empty elements of node are belong to14:the same class.15:Output: A decision tree.

## 5. Simulation Results

For the purpose of proving the effectiveness of the proposed method as to aerial target state prediction and intention recognition, the simulation data is given in Table 1, Table 2 and Table 3.

Table 1 is the real-time numeric data of an aerial target. In this paper, a typical scenario of the target with attack intention is considered. It is assumed that the azimuth basically remains unchanged, the distance between target and UCAV is reduced gradually, the velocity of target is increased to a stable value, and the heading angle fluctuates in a certain range and the height is decreased to a narrow range.

Table 2 is the real-time nonnumeric data of aerial target.

Table 3 is the historical data of past air combat.

In Table 3, “*” denotes that the value of the corresponding air combat attribute is unknown.

In order to utilize the real-time data to train the LSTM networks, we choose the size of time window p=4. The number of cells of LSTM networks is 12.

Furthermore, in order to demonstrate the efficiency and feasibility of the proposed state prediction approach, ARIMA method [6] is compared with LSTM networks method in this part. We choose the real-time numeric data of time 1 to time 14 as the training set, time 15 as the test set to test the predicted results. The simulation results are shown in Table 4 and Table 5.

From the performance comparisons of ARIMA and LSTM networks, prediction accuracy of LSTM networks is significantly better than ARIMA. For example, the predicted result of ARIMA in the azimuth is totally wrong. On the other hand, the training time using LSTM networks is one of the drawbacks but the state prediction can be parallelized on military systems with high performance computer systems.

The output of LSTM networks and the predicted state data of target are shown in Table 6.

In Table 6, the predicted values of azimuth, distance, velocity, heading angle and height are obtained by the LSTM networks.

In accordance with fuzzy inference, a fuzzy incomplete decision table of historical numeric data part is generated as Table 7.

From Table 3 and Table 7, the partitions of all condition attributes can be obtained
$$\begin{array}{c} \begin{array}{ccc} U/A=A_{E},A_{S},A_{N},A_{W}, & U/D=D_{S},D_{M},D_{L}, & U/V=V_{S},V_{M},V_{F},\\ U/Ha=Ha_{S},Ha_{M},Ha_{L}, & U/H=H_{L},H_{M},H_{H}. & U/Ars={Ars}_{0},{Ars}_{1},\\ U/Mrs={Mrs}_{0},{Mrs}_{1}, & U/Ds={Ds}_{0},{Ds}_{1}, & U/Dds={Dds}_{0},{Dds}_{1}. \end{array} \end{array}$$
where $A_{E}$, $A_{S}$, $A_{N}$ and $A_{W}$ are the partitions of East, South, North and West in condition attribute azimuth, $D_{S}$, $D_{M}$ and $D_{L}$ are the partitions of Short, Medium and Long in condition attribute distance, $V_{S}$, $V_{M}$ and $V_{F}$ are the partitions of Small, Medium and Fast in condition attribute velocity, ${Ha}_{S}$, ${Ha}_{M}$ and ${Ha}_{L}$ are the partitions of Short, Medium and Large in condition attribute heading angle, $H_{L}$, $H_{M}$ and $H_{H}$ are the partitions of Low, Medium and High in condition attribute height, ${Ars}_{0}$ and ${Ars}_{1}$ are the partitions of 0 and 1 in condition attribute air-to-air radar status, ${Mrs}_{0}$ and ${Mrs}_{1}$ are the partitions of 0 and 1 in condition attribute marine radar status, ${Ds}_{0}$ and ${Ds}_{1}$ are the partitions of 0 and 1 in condition attribute disturbing state, ${Dds}_{0}$ and ${Dds}_{1}$ are the partitions of 0 and 1 in condition attribute disturbed state.

Then, we have
$$\begin{array}{c} \begin{array}{c} A_{E}=\left\{1,2,3,8,9,10,14,15,17,20,21,22\right\},A_{S}=\left\{2,4,5,6,9,13,14,15,16,20,22,23\right\},\\ A_{N}=\left\{2,7,9,14,15,18,20,23\right\},A_{W}=\left\{2,9,11,12,14,15,19,20,22\right\}.\\ D_{S}=\left\{2,3,4,12,13,14,15,17,19\right\},D_{M}=\left\{1,3,4,7,8,9,11,16,18,19,20,22,23\right\},\\ D_{L}=\left\{3,4,5,6,10,19,21\right\}.\\ V_{S}=\left\{8,9,10,18,22\right\},V_{M}=\left\{8,11,12,16,17,19,20,21,23\right\},\\ V_{F}=\left\{1,2,3,4,5,6,7,8,13,14,15\right\}.\\ Ha_{S}=\left\{1,2,3,4,6,16,17,19,20,22,23\right\},{Ha}_{M}=\left\{5,6,7,8,9,10,11,12,16,19,20,22,23\right\},\\ {Ha}_{L}=\left\{6,13,14,15,16,18,19,20,21,22,23\right\}.\\ H_{L}=\left\{1,2,3,4,9,11,12,13,14,15,21,23\right\},H_{M}=\left\{2,5,6,7,8,9,10,16,17,19,21,22,23\right\},\\ H_{H}=\left\{2,9,18,20,21,23\right\}.\\ {Ars}_{0}=\left\{11,15,18,19,20,21,23\right\},\\ {Ars}_{1}=\left\{1,2,3,4,5,6,7,8,9,10,11,12,13,14,15,16,17,19,22,23\right\}.\\ {Mrs}_{0}=\left\{1,2,3,10,14,16,17,19,20,21\right\},\\ {Mrs}_{1}=\left\{2,3,4,5,6,7,8,9,10,11,12,13,14,15,16,18,20,21,22,23\right\}.\\ {Ds}_{0}=\left\{2,5,7,8,9,11,12,18,19,20,21,22\right\},\\ {Ds}_{1}=\left\{1,2,3,4,5,6,10,12,13,14,15,16,17,18,22,23\right\}.\\ {Dds}_{0}=\left\{1,2,3,5,6,7,8,9,11,12,16,17,18,19,20,21,22,23\right\},\\ {Dds}_{1}=\left\{3,4,5,6,7,9,10,11,13,14,15,17,22\right\}. \end{array} \end{array}$$

According to (9), the decision support degree of each condition attribute with decision attribute (intention *I*) can be calculated, which is given by
$$\begin{array}{c} \begin{array}{ccc} DSD(A,I)=0.5836, & DSD(D,I)=0.6154, & DSD(V,I)=0.6538,\\ DSD(Ha,I)=0.5746, & DSD(H,I)=0.5938, & DSD(Ars,I)=0.3493,\\ DSD(Mrs,I)=0.3561, & DSD(Ds,I)=0.4933, & DSD(Dds,I)=0.5219. \end{array} \end{array}$$

Hence, the condition attribute velocity is selected to be split firstly.

Back to Table 3, sort the endpoints of velocity elements in ascending order and delete the repetitive endpoints, we have
$$\begin{array}{c} 110,120,140,150,170,200,210,220,230,240,250,270,280,290,300,315,320,330 \end{array}$$

The alternative cutpoints are
$$\begin{array}{c} 115,130,145,160,185,205,215,225,235,245,260,275,285,295,307.5,317.5,325 \end{array}$$

The corresponding information entropy of partitioning can be obtained with (15) and (16), which is shown as follows.
$$\begin{array}{c} 2.0353,1.9372,1.8260,1.8132,1.5516,1.5516,1.5516,1.5688,1.5688,\\ 1.4681,1.4093,1.4093,1.4093,1.5240,1.6618,1.8064,1.8567. \end{array}$$
where the minimum information entropy of partitioning is 1.4093, and the first corresponding alternative cutpoint is 260.0. Therefore, the cutpoint 260.0 is chosen as the optimal cutpoint.

Repeat the process until the decision tree is generated, which is shown in Figure 12.

Finally, after inputting the predicted state data into the decision tree, the intention of the target is recognized as attack, and it is consistent with the simulation scenario.

From the LSTM networks predicted results, the predicted model can effectively predict the variation trend of real-time numeric data. The suitable time window can also solve the lack of training data problem. In addition, the generated decision tree based on the uncertain and incomplete priori knowledge indicates that the proposed method is also applicable to deal with the uncertainty and incompleteness in air combat. Thus, the method presented is practical and effective in intention prediction of aerial targets.

## 6. Conclusions

As the basis of future air combat autonomous decision-making systems, intention prediction of aerial target makes a positive contribution to unmanned systems. In this paper, a state prediction and intention recognition method is developed. The future state information of a target is predicted based on LSTM networks from real-time series data, and the uncertainty and incompleteness of priori knowledge is expressed as an interval-valued number and null value in an air combat information system. To generate a decision tree based on the uncertain and incomplete historical data, a decision support degree is applied for node splitting order selection and a split criterion is determined by the information entropy of partitioning. Then, the target intention is obtained by the predicted data and the built decision tree. The simulation result shows that the deserved algorithm can predict state and recognize intention of an aerial target. Therefore, the proposed method could handle the air combat information system, in which there is a great deal of uncertainty and incompleteness. However, the relationship between target state and intention needs to be further explored. For future work, air combat attack decision-making based on target intention is worth considering. 

## Figures and Tables

**Figure 1 entropy-22-00279-f001:**
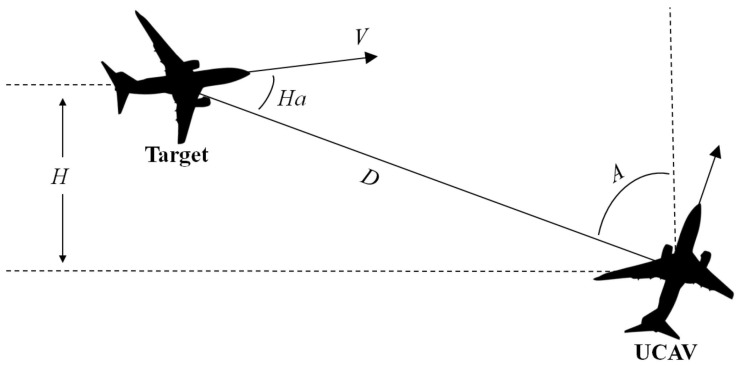
Air combat situation diagram.

**Figure 2 entropy-22-00279-f002:**
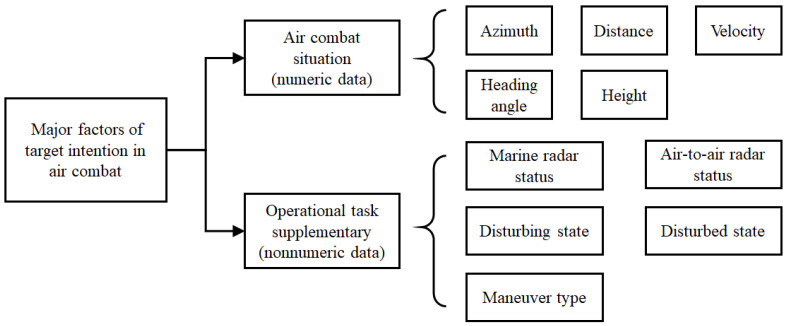
The tree chart of target intention characteristic description in air combat.

**Figure 3 entropy-22-00279-f003:**
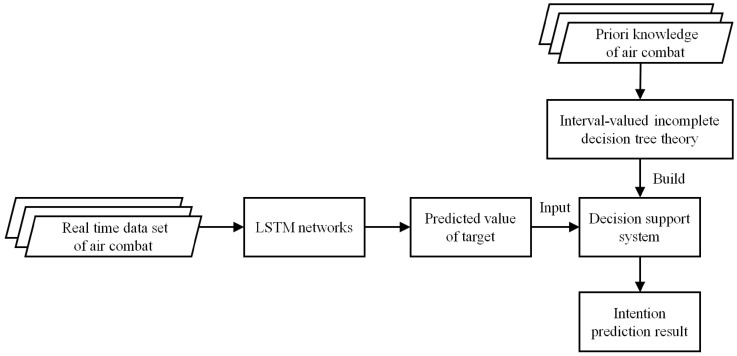
The intention prediction system diagram.

**Figure 4 entropy-22-00279-f004:**
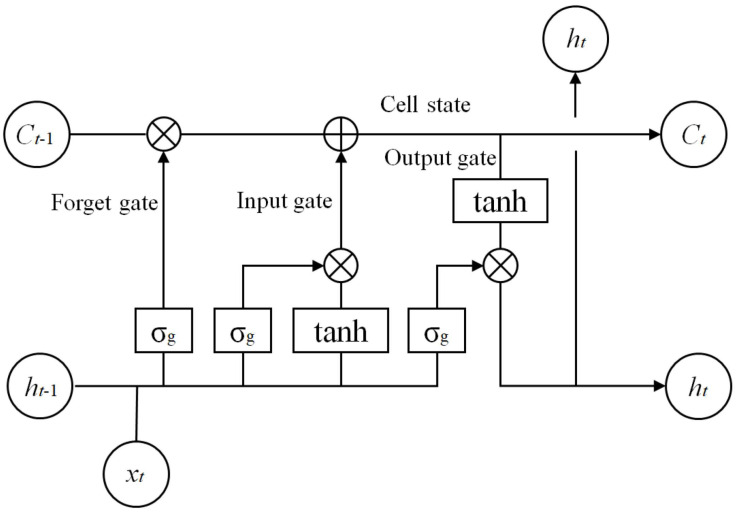
The structure of LSTM networks.

**Figure 5 entropy-22-00279-f005:**
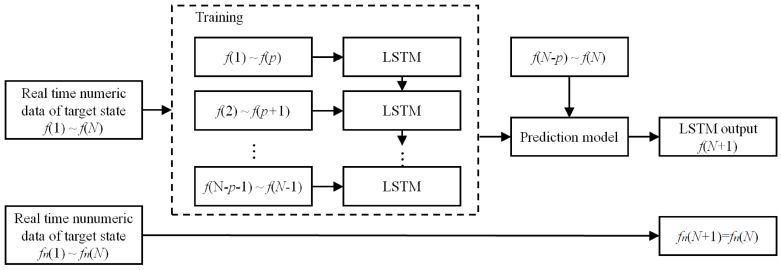
The structure of aerial target state prediction.

**Figure 6 entropy-22-00279-f006:**
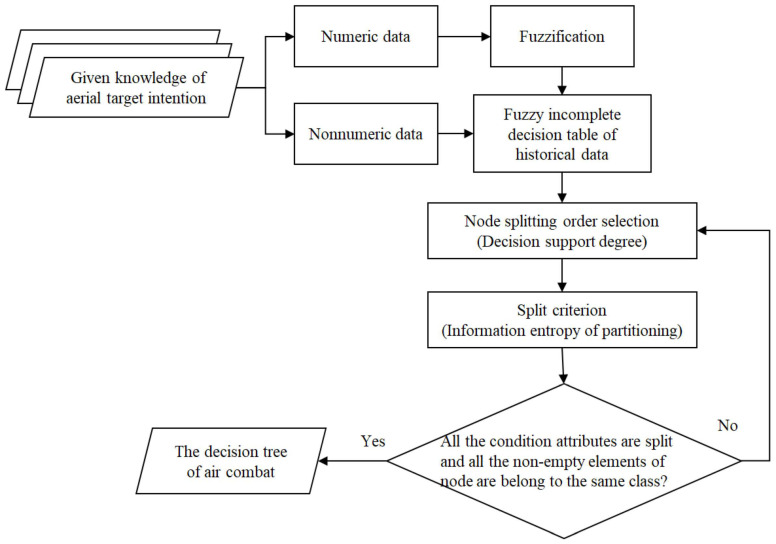
The structure of decision tree generation for aerial target intention recognition.

**Figure 7 entropy-22-00279-f007:**
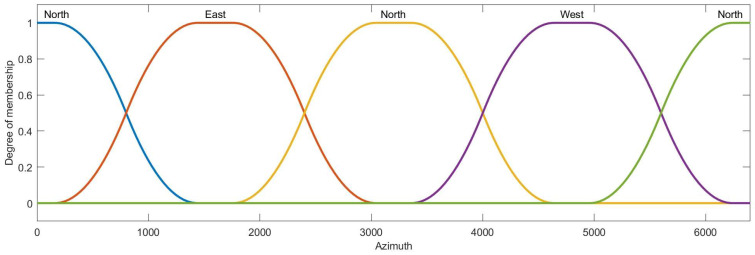
The membership function curve of azimuth.

**Figure 8 entropy-22-00279-f008:**
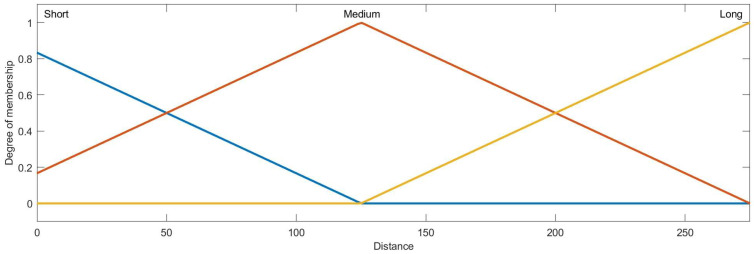
The membership function curve of distance.

**Figure 9 entropy-22-00279-f009:**
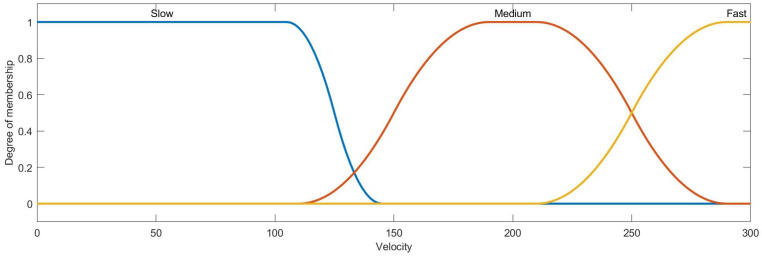
The membership function curve of velocity.

**Figure 10 entropy-22-00279-f010:**
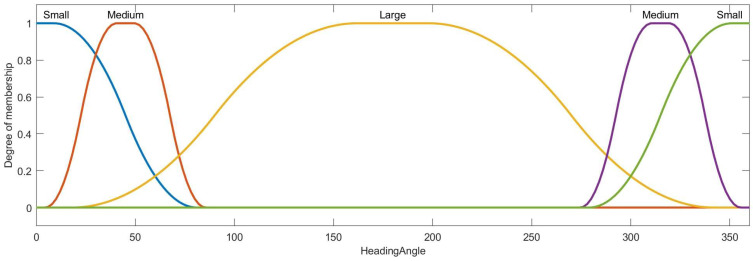
The membership function curve of heading angle.

**Figure 11 entropy-22-00279-f011:**
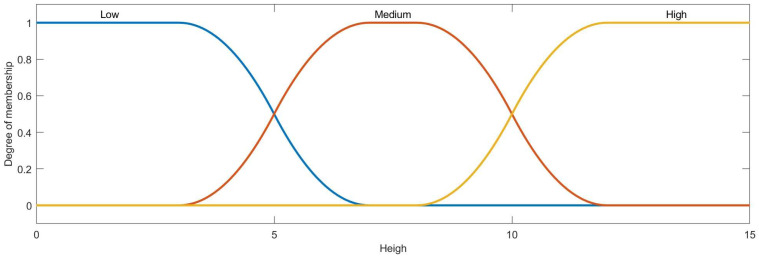
The membership function curve of height.

**Figure 12 entropy-22-00279-f012:**
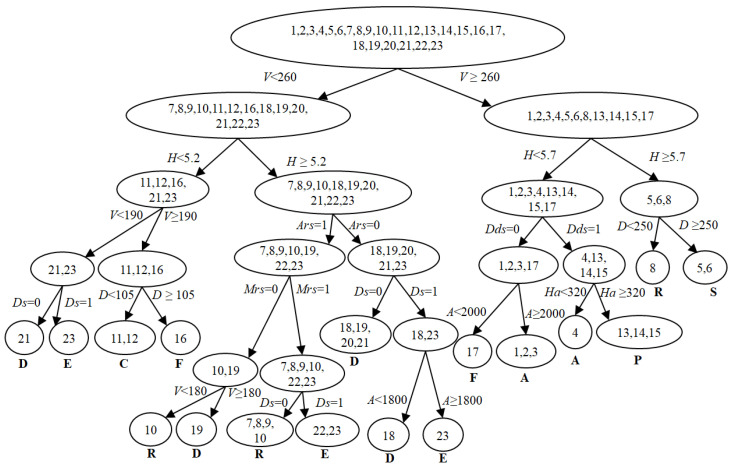
The decision tree of uncertain and incomplete priori knowledge in air combat.

**Table 1 entropy-22-00279-t001:** Real time numeric data of aerial target.

Time	Azimuth (mil)	Distance (km)	Velocity (m/s)	Heading Angle $\left({}^{\circ }\right)$	Heigh (km)
1	2230.0	310.0	220.0	12.0	15.8
2	2245.0	297.0	242.0	11.0	13.2
3	2257.0	291.0	228.0	14.0	11.7
4	2300.0	280.0	241.0	13.0	10.1
5	2364.0	267.0	255.0	10.0	8.6
6	2413.0	251.0	267.0	8.0	7.4
7	2467.0	235.0	263.0	5.0	6.0
8	2489.0	214.0	285.0	14.0	5.2
9	2488.0	199.0	274.0	4.0	4.5
10	2514.0	178.0	286.0	7.0	3.7
11	2516.0	154.0	293.0	11.0	3.2
12	2524.0	138.0	285.0	10.0	3.1
13	2517.0	120.0	284.0	9.0	2.8
14	2536.0	97.0	292.0	10.0	2.5
15	2528.0	75.0	291.0	6.0	2.6

**Table 2 entropy-22-00279-t002:** Real time nonnumeric data of aerial target.

Time	Air-to-Air Radar Status	Marine Radar Status	Disturbing State	Disturbed State
15	1	0	1	0

**Table 3 entropy-22-00279-t003:** Given knowledge of air target intention. (the bold is used to mark off decision attribute and condition attributes).

Index	Azimuth (mil)	Distance (km)	Velocity (m/s)	Heading Angle (${}^{\circ }$)	Height (km)
1	[2200.0,2300.0]	[100.0,110.0]	[300.0,320.0]	[10.0,30.0]	[4.0,5.0]
2	∗	[45.0,55.0]	[320.0,330.0]	[320.0,350.0]	∗
3	[2200.0,2300.0]	∗	[300.0,330.0]	[30.0,40.0]	[2.0,2.5]
4	[2800.0,3000.0]	∗	[270.0,290.0]	[330.0,350.0]	[3.6,4.0]
5	[2800.0,2850.0]	[260.0,290.0]	[315.0,320.0]	[80.0,90.0]	[7.7,8.0]
6	[2800.0,2850.0]	[240.0,260.0]	[300.0,315.0]	∗	[6.7,7.2]
7	[750.0,810.0]	[180.0,190.0]	[150.0,170.0]	[60.0,80.0]	[6.0,6.5]
8	[820.0,830.0]	[180.0,185.0]	∗	[70.0,90.0]	[6.5,7.7]
9	∗	[160.0,180.0]	[110.0,120.0]	[40.0,60.0]	∗
10	[820.0,860.0]	[200.0,220.0]	[120.0,140.0]	[50.0,70.0]	[5.4,6.0]
11	[4000.0,4100.0]	[50.0,60.0]	[210.0,220.0]	[60.0,90.0]	[3.4,3.6]
12	[4020.0,4050.0]	[35.0,45.0]	[210.0,250.0]	[70.0,100.0]	[2.0,2.6]
13	[2600.0,2800.0]	[30.0,40.0]	[280.0,300.0]	[140.0,160.0]	[2.0,3.0]
14	∗	[0,20.0]	[300.0,320.0]	[160.0,180.0]	[2.0,2.4]
15	∗	[25.0,35.0]	[290.0,300.0]	[210.0,240.0]	[0,3.0]
16	[2400.0,2500.0]	[150.0,160.0]	[230.0,240.0]	∗	[4.4,5.0]
17	[1700.0,1800.0]	[50.0,60.0]	[300.0,320.0]	[20.0,30.0]	[5.0,5.4]
18	[600.0,800.0]	[160.0,180.0]	[120.0,140.0]	[150.0,170.0]	[9.6,10.6]
19	[5000.0,5200.0]	∗	[220.0,240.0]	∗	[8.0,9.0]
20	∗	[150.0,160.0]	[230.0,240.0]	∗	[10.0,10.6]
21	[810.0,860.0]	[200.0,220.0]	[200.0,220.0]	[200.0,210.0]	∗
22	∗	[180.0,200.0]	[120.0,150.0]	∗	[6.0,7.2]
23	[2800.0,2900.0]	[160.0,180.0]	[140.0,170.0]	∗	∗
**Index**	**Air-to-air** **radar status**	**Marine radar** **status**	**Disturbing** **state**	**Disturbed** **state**	**Intention**
1	1	0	1	0	A
2	1	∗	∗	0	A
3	1	∗	1	∗	A
4	1	1	1	1	A
5	1	1	∗	∗	S
6	1	1	1	∗	S
7	1	1	0	∗	R
8	1	1	0	0	R
9	1	1	0	∗	R
10	1	∗	0	1	R
11	∗	1	0	∗	C
12	1	1	∗	0	C
13	1	1	1	1	P
14	1	∗	1	1	P
15	∗	1	1	1	P
16	1	∗	1	0	F
17	1	0	1	∗	F
18	0	1	∗	0	D
19	∗	0	0	0	D
20	0	∗	0	0	D
21	0	∗	0	0	D
22	1	1	∗	∗	E
23	∗	1	1	0	E

**Table 4 entropy-22-00279-t004:** The state predicted value at time 15 of ARIMA and LSTM networks.

Predicted Value					
Method	Azimuth (mil)	Distance (km)	Velocity (m/s)	Heading Angle (${}^{\circ }$)	Height (km)
ARIMA	2996.0	65.7	297.7	9.6	1.7
LSTM networks	2512.5	71.8	294.3	9.3	2.6

**Table 5 entropy-22-00279-t005:** The performance comparisons of ARIMA and LSTM networks.

Error					
Method	Azimuth	Distance	Velocity	Heading Angle	Height
ARIMA	468.0	9.3	6.7	3.6	0.9
LSTM networks	15.5	3.2	3.3	3.3	0
**Test time (s)**					
**Method**	**Azimuth**	**Distance**	**Velocity**	**Heading angle**	**Height**
ARIMA	3.13	2.20	2.34	3.77	1.69
LSTM networks	2.42	2.11	2.65	0.86	0.92

**Table 6 entropy-22-00279-t006:** The predicted state data of target.

Time	Azimuth (mil)	Distance (km)	Velocity (m/s)	Heading Angle (${}^{\circ }$)	Height (km)
16	2526.0	60.7468	284.9901	8.4227	2.4236
**Time**	**Air-to-air** **radar status**	**Marine radar** **status**	**Disturbing** **state**	**Disturbed** **state**	
16	1	0	1	0	

**Table 7 entropy-22-00279-t007:** The fuzzy incomplete decision table of historical numeric data part. (the bold is used to mark off decision attribute and condition attributes).

Index	Azimuth (mil)	Distance (km)	Velocity (m/s)	Heading Angle (${}^{\circ }$)	Height (km)	Intention
1	East	Medium	Fast	Small	Low	A
2	∗	Short	Fast	Small	∗	A
3	East	∗	Fast	Small	Low	A
4	South	∗	Fast	Small	Low	A
5	South	Long	Fast	Medium	Medium	S
6	South	Long	Fast	∗	Medium	S
7	North	Medium	Medium	Medium	Medium	R
8	East	Medium	∗	Medium	Medium	R
9	∗	Medium	Slow	Medium	∗	R
10	East	Long	Slow	Medium	Medium	R
11	West	Medium	Medium	Medium	Low	C
12	West	Short	Medium	Medium	Low	C
13	South	Short	Fast	Large	Low	P
14	∗	Short	Fast	Large	Low	P
15	∗	Short	Fast	Large	Low	P
16	South	Medium	Medium	∗	Medium	F
17	East	Short	Medium	Small	Medium	F
18	North	Medium	Slow	Large	High	D
19	West	∗	Medium	∗	Medium	D
20	∗	Medium	Medium	∗	High	D
21	East	Long	Medium	Large	∗	D
22	∗	Medium	Slow	∗	Medium	E
23	South	Medium	Medium	∗	∗	E

“∗” denotes that the value of the corresponding air combat attribute is unknown.

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
