# Peer review of "Information Entropy-Based Intention Prediction of Aerial Targets under Uncertain and Incomplete Information"

_entropy, 2020, doi:10.3390/e22030279_

Round 1
Reviewer 1 Report
Dear editor,
Thank you for inviting me for review the paper “Information Entropy Based Intention Prediction of Aerial Target under Uncertain and Incomplete Information”. The authors presented military decision support system (DSS) for prediction and intention recognition. The paper is well presented. The methodology has a great potential in decision making process especially for military field, so this topic in important for investigation. Results are good. The paper is relevant. However, the author(s) need to consider the following points as limitation or further scope for refining the paper:
- Abstract should be rewritten. You should present in the abstract: Purpose; Design/methodology/approach; Findings; Practical implications; Originality/value. There is no information about FLS used in the paper.
- Introduction section should be revised. Please clearly summarise what specific advantages brings your approach. What are advantages of it? Try to specific that and build a case for your research (focus on novelty). This should be presented in at least one paragraph.
- Please clarify what is the novelty of your work.
- Gaps in the literature could be mentioned more clearly in the results section.
- Literature review (LR) is important for showing that authors are familiar with relevant research literature for proposed field. There is no LR in this paper. I know that literature with military DSS is limited, but there are several recent published papers from that field. For example, Karavidic and Projovic, 2018, presented interesitng application of rough set theory for decision making in army forces (A multi-criteria decision-making (MCDM) model in the security forces operations based on rough sets. Decision Making: Applications in Management and Engineering), also Petrovic and Kankaras, 2018. presented hybrid MCDM model for the selection and evaluation of criteria for selecting an aircraft for the protection of air traffic (DEMATEL-AHP multi-criteria decision making model for the selection and evaluation of criteria for selecting an aircraft for the protection of air traffic. Decision Making: Applications in Management and Engineering).
- Results are concise and well presented.
- Summaries the conclusion and write what you already achieved in this research. Add more future scope. Add limitations of proposed algorithm.
I must iterate that the authors presented very quality research.
Regards,
Reviewer
Reviewer 2 Report
This manuscript proposed the state prediction and intention recognition of aerial target by using LSTM method under with uncertain and incomplete information. It is very interesting issue and well written. But, I suggest to improve the quality of paper for publishing.
- It should be added the comparison of performance results such as accuracy and learning and test time with LSTM and others (ARIMA, etc.).
- Experimental environments should be more explained such as learning data size. This method can be applied to practical application? If the author's answer is YES, they should be discussed about the practical issue. Because it tested with the simulation data.
- It should be added “failure case”. Why LSTM method is better than ARIMA to solve this problem?
Reviewer 3 Report
The paper addresses the very active field of research: training of RNNs with LSTM units, with many and diverse applications. A particular focus is on Target Intention Recognition Based on Decision Tree and Information Entropy.
The English has to be improved throughout the manuscript. The text also needs improvement to help understanding for the non-specialist reader who may not be familiar with the topic.
Abstract, lines 6 - 7, as well as line 18
"Then, the target intention is obtained by the predicted data and the built decision tree."
Please explain briefly what is meant by 'intention'. (I know this is explained later - in section 2 - but it would help the reader understanding.)
Also better English would be: "Then, the target intention is obtained from the predicted data by applying the built decision tree." Please confirm I've understood.
Line 23. Reference [2] is not yet published. Can the authors therefore confirm that the present manuscript contains significantly new material to warrant publication, compared with the work of [2].
Line 27. Reference [3] concerns a different application to the present work. Please explain to the reader by modifying this sentence: "online lane change intention prediction", for example by adding the words "...in road vehicle driving". Is the assumption, therefore, that flying an aircraft and driving a car are sufficiently similar for useful analogies to be drawn?
At line 37, mention is also made to maritime trafic, in reference [8]. At line 47 is made mention to activity recognition which in [14] is in the context of wearble sensing. Speech and handwriting recognition are mentioned amongst many other additional applications on the Wikipedia page on LSTM [https://en.wikipedia.org/wiki/Long_short-term_memory]. Are all of these applications sufficiently similar? Is activity recognition the key concept?
Line 42. Please explain the abbreviation LSTM = long short-term memory here (as well as in the abstract).
Line 43. Traditional RNNs need to be referenced. Do any the references [1],...,[8] given so far mention RNNs? If not, please give another reference to them - such as current reference [22].
Line 43. The authors should add a few words to explain that "exploding and vanishing gradient problem" does not have to do specifically with their air combat application, but is about finite precision computation (rather than exploding aircraft), if I have understood correctly.
Lines 94, 147 and 198. Could one add to the text: that "nonnumeric data" lie on a nominal scale?
Line 125 and later occurences of sigma: Please write sigma sub g to denote explicitly the sigmoid function [Wikipedia page on LSTM], to clearly distinguish it from for example sigma sub c = the hyperbolic tangent function.
Line 129 could be rewritten for clarity: Replace "If ft = 1, completely keep the information of ht and xt. Otherwise, ft = 0 means completely gotten rid of the information" with "If ft = 1, information about ht and xt is retained completely, while ft = 0 means completely losing the information". It is important to indicate that there is a complete (continuous?) range - not just two end-points, if I've understood correctly?
Line 152 The text: "predicted model" should read "prediction model".
Line 157 "Because of the incompleteness and the uncertainty of air combat, it is hard to extract the rules from historical data". Apart from a statistics-based approach, would the authors consider also belief or plausibility when formulating membership functions [e.g. C de Silva 1995, Intelligent control: Fuzzy logics applications, ISBN 0-8493-7982-2; p. 144 of Pendrill 2019 Quality Assured Measurement – Unification across Social and Physical Sciences, Springer Series in Measurement Science and Technology, ISBN: 978-3-030-28695-8 (e-book), https://doi.org/10.1007/978-3-030-28695-8; and [27]]?
Line 172 The text "As is all known" should read "As is well known".
Line 205 "The fuzzy inference is used to solve this problem". Is it obvious that the fuzzy approach captures all essentialities? Fuzziness might be good at treating dispersion (scatter), but how well does it deal with bias, i.e. systematic error in location? See Pendrill 2019, p. 56.
Line 211 Figure 11. What determines the widths of the areas of transition from one category to another in each of the membership function curves? Is it uncertainty or perhaps the finite resolution of each sensor, or something else?
Line 225 "The information entropy of selected condition attribute ..." It is known that entropy according to the expression eq. 15 is not a complete metric - there are several alternatives: see Pendrill 2019, p 169. Have the authors considered the implications of this limitation in entropy as a metric for their current research?
Line 242 Table 1. Apart from the last column ('Height'), the data includes a first decimal which is always zero: '0'. Is that a significant digit, or can it be simply omitted? Please, also, indicate the measurement uncertainties in each set of data and eliminate non-significant digits.
Line 247 Table 4. I guess that few of the digits quoted in for example the distance "60.7468 km" are significant. Therefore, please indicate the measurement uncertainties in each set of data and eliminate non-significant digits.
Round 2
Reviewer 1 Report
I read through the paper and thought most of my concerns were well addressed, thus it was acceptable from my side.